# Classification of the human phox homology (PX) domains based on their phosphoinositide binding specificities

Mintu Chandra[1], Yanni K.-Y. Chin[1], Caroline Mas[1,6], J. Ryan Feathers[2], Blessy Paul[1], Sanchari Datta[2], Kai-En Chen [1], Xinying Jia [3], Zhe Yang[4], Suzanne J. Norwood[1], Biswaranjan Mohanty[5], Andrea Bugarcic[1], Rohan D. Teasdale[1,4], W. Mike Henne[2], Mehdi Mobli [3] & Brett M. Collins[1]

Phox homology (PX) domains are membrane interacting domains that bind to phosphatidylinositol phospholipids or phosphoinositides, markers of organelle identity in the endocytic system. Although many PX domains bind the canonical endosome-enriched lipid PtdIns3P, others interact with alternative phosphoinositides, and a precise understanding of how these specificities arise has remained elusive. Here we systematically screen all human PX domains for their phospholipid preferences using liposome binding assays, biolayer interferometry and isothermal titration calorimetry. These analyses define four distinct classes of human PX domains that either bind specifically to PtdIns3P, non-specifically to various di- and tri-phosphorylated phosphoinositides, bind both PtdIns3P and other phosphoinositides, or associate with none of the lipids tested. A comprehensive evaluation of PX domain structures reveals two distinct binding sites that explain these specificities, providing a basis for defining and predicting the functional membrane interactions of the entire PX domain protein family.

[1] Institute for Molecular Bioscience, The University of Queensland, St. Lucia, QLD 4072, Australia. [2] Department of Cell Biology, University of Texas Southwestern Medical Center, Dallas, TX 75390, USA. [3] Centre for Advanced Imaging and School of Chemistry and Molecular Biology, The University of Queensland, St. Lucia, QLD 4072, Australia. [4] School of Biomedical Sciences, Faculty of Medicine, The University of Queensland, St. Lucia, QLD 4072, Australia. [5] Medicinal Chemistry, Monash Institute of Pharmaceutical Sciences, Monash University, 381 Royal Parade, Parkville 3052 VIC, Australia. [6] Present address: Integrated Structural Biology Grenoble, Grenoble, France. These authors contributed equally: Yanni K.-Y. Chin, Caroline Mas. Correspondence and requests for materials should be addressed to B.M.C. (email: b.collins@imb.uq.edu.au)

The phosphoinositides or phosphatidylinositol phospholipids (PtdIns*P*s) are, along with the Rab family of small GTPase proteins, among the most important markers of organelle identity within the secretory and endocytic system. Phosphoinositides are differentially phosphorylated at the D3, D4, and D5 positions of the 6-carbon inositol ring by phosphoinositide kinases and phosphatases and provide both specific membrane anchors for recruiting peripheral membrane proteins, as well as substrates for generating other lipids and second messengers, such as $Ins(1,4,5)P_3$[1]. A variety of protein domains have been discovered with specific PtdIns*P*-binding properties, and these are found in proteins with diverse functions in membrane trafficking, cell signaling, or membrane-associated scaffolding roles[2–4]. Examples include pleckstrin homology (PH), fab1/yotb/vac1/eea1 (FYVE), and C2 domains. Several well-characterized domains are now used as tools for imaging the cellular locations of different PtdIns*P*s, such as plasma membrane-enriched $PtdIns(4,5)P_2$ by the PH domain of PLCδ, transiently produced $PtdIns(3,4,5)P_3$ by the PH domain of Grp1, and the endosomal lipid PtdIns3*P* by the FYVE domain of EEA1. This has all led to the concept of a phosphoinositide code that is recognized by these various PtdIns*P* effectors[3], although it is important to understand that many of these lipid-effector interactions are of relatively low affinity, and are often conditional on coordinated binding or coincident detection of other localization signals, such as the presence of Rabs or other lipids[4].

The Phox homology (PX) domain was first identified in the NADPH phagocyte oxidase complex subunits p40[phox] and p47[phox][5], and has since been found in proteins in every eukaryotic species from yeast to human[6–8]. The human genome encodes 49 proteins that possess PX domains, many of which are termed sorting nexins (SNXs), that all play diverse roles in signaling, trafficking, and membrane homeostasis. The PX domain was established early on as a phosphoinositide-binding domain, and shown to bind to the endosomal lipid PtdIns3*P* based on studies of yeast Vam7, human SNX3, and p40[phox][9–16]. Subsequent work showed that the majority of yeast PX domains had preferential affinity for PtdIns3*P*[17], and studies since have shown that specific interactions with PtdIns3*P* are essential for cellular localization of many PX domain proteins to the outer leaflet of early endosomes in various organisms[7,8]. Nonetheless, affinities of PX domains for a variety of other phosphoinositides and lipids have also been reported[7], including p47phox[18] and PI3KC2α[19].

Despite relatively low overall sequence homology, PX domains all possess the same core fold, consisting of three antiparallel β-strands (β1-β3), followed by three α-helices (α1–α3)[8]. An extended sequence traverses the protein between helices α1 and α2, termed the PPK loop as it generally contains a conserved ΨPxxPxK motif (Ψ = large aliphatic amino acids V, I, L, and M). Side chains of residues from the β3 strand, α1 helix and PPK loop together form a binding pocket for the headgroup of the canonical lipid PtdIns3*P*. Several crystal structures have now defined the molecular details of the stereo-specific recognition of PtdIns3*P*, and mutations in the side-chains that bind the PtdIns3*P* headgroup invariably cause dissociation of these PX domain proteins from endosomal compartments. Notwithstanding this knowledge, the mechanism(s) by which alternative or noncanonical phosphoinositides bind to PX domains, the range of different phosphoinositides that associate with PX domains, and the significance of these interactions for cellular localization remain unknown. One difficulty is that many different methods have been used to measure PX domain interactions with membranes, often with either different or conflicting results (see Supplementary Table 1). Given the importance of PX domain proteins in normal cell physiology, in diseases as diverse as

Alzheimer's, cancer, and pathogen invasion, a clearer understanding of PX domain membrane affinities is critical.

In this study, we have purified over three-quarters of the human PX domains and performed a systematic and comprehensive analysis of their phosphoinositide-binding properties using qualitative liposome pelleting assays and quantitative isothermal titration calorimetry (ITC) and biolayer interferometry (BLItz) biophysical measurements. These studies resolve four distinct categories of PX domains; those that bind specifically to the canonical lipid PtdIns3*P*, those that bind other phosphoinositides as well as PtdIns3*P*, those that bind other lipids but not PtdIns3*P*, and those that do not appear to bind membrane lipids significantly at all. An analysis of PX domain sequences and structures, including several new crystallographic and NMR structures, suggest that a common determinant of noncanonical lipid binding is the presence of a His or Tyr side chain within helix α1 along with adjacent basic residues. Using structure-guided mutagenesis, we demonstrate the general role of this secondary His/Tyr-containing site in noncanonical lipid binding and have verified the in vitro observations for several family members using a yeast cell-based membrane recruitment assay. Altogether, our biochemical, structural, and cellular studies provide a thorough description of the membrane-binding specificities of the human PX domain family, with potential implications for their roles in signal transduction and membrane trafficking.

## Results

**Classifying the human PX domain PtdIns*P* specificity.** As summarized in Supplementary Table 1, many previous reports have described interactions of human PX domain proteins with a variety of different phosphoinositide lipids, as well as other lipids such as phosphatidylserine (PS) and phosphatidic acid (PA) that may promote recruitment to various cellular organelles and membrane domains. Our aim was to perform parallel comparisons of as many human PX domains as possible, using qualitative and quantitative approaches to measure binding to both membrane mimetics and soluble phosphoinositide headgroups. We first cloned 46 of the 49 human PX domains with the N-terminal GST fusion tags and expressed them in bacteria, followed by purification by glutathione affinity chromatography and removal of the GST-tag by thrombin cleavage. Our strict criterion for further analysis was that the PX domains could be isolated as predominantly single and homogeneous species via gel filtration. Of the 46 human PX domains cloned, we successfully purified 39 for further study (Supplementary Figure 1; Supplementary Table 2).

We next assessed the binding of the human PX domain containing proteins to membranes containing phosphoinositides using qualitative liposome pelleting assays (Figs. 1, 2a). None of the tested proteins showed nonspecific pelleting in buffer or with control PC/PE membranes. As expected a number of PX domains showed exclusive interaction with liposomes containing PtdIns3*P* (SNX3, SNX7, SNX19, SNX17, SNX27, p40phox, PXK, and SNX16), with SNX11 and SNX12 showing a slight preference for this endosome-enriched lipid. The somewhat surprising result was that the PX domains that bound selectively to the canonical PtdIns3*P* were in the minority. More commonly the PX domains either showed a range of lipid-binding preferences or did not bind to any of the membranes under the conditions tested. The nonbinding proteins included SNX5, SNX6, SNX32, and SNX14, and these are unlikely to be false negatives due to poor folding as all four domains are otherwise functional in previous protein-binding or structural studies[20,21]. In our pelleting experiments, we have also used Folch fraction brain extracted lipids, as we find that they provide a general indication of broader lipid binding

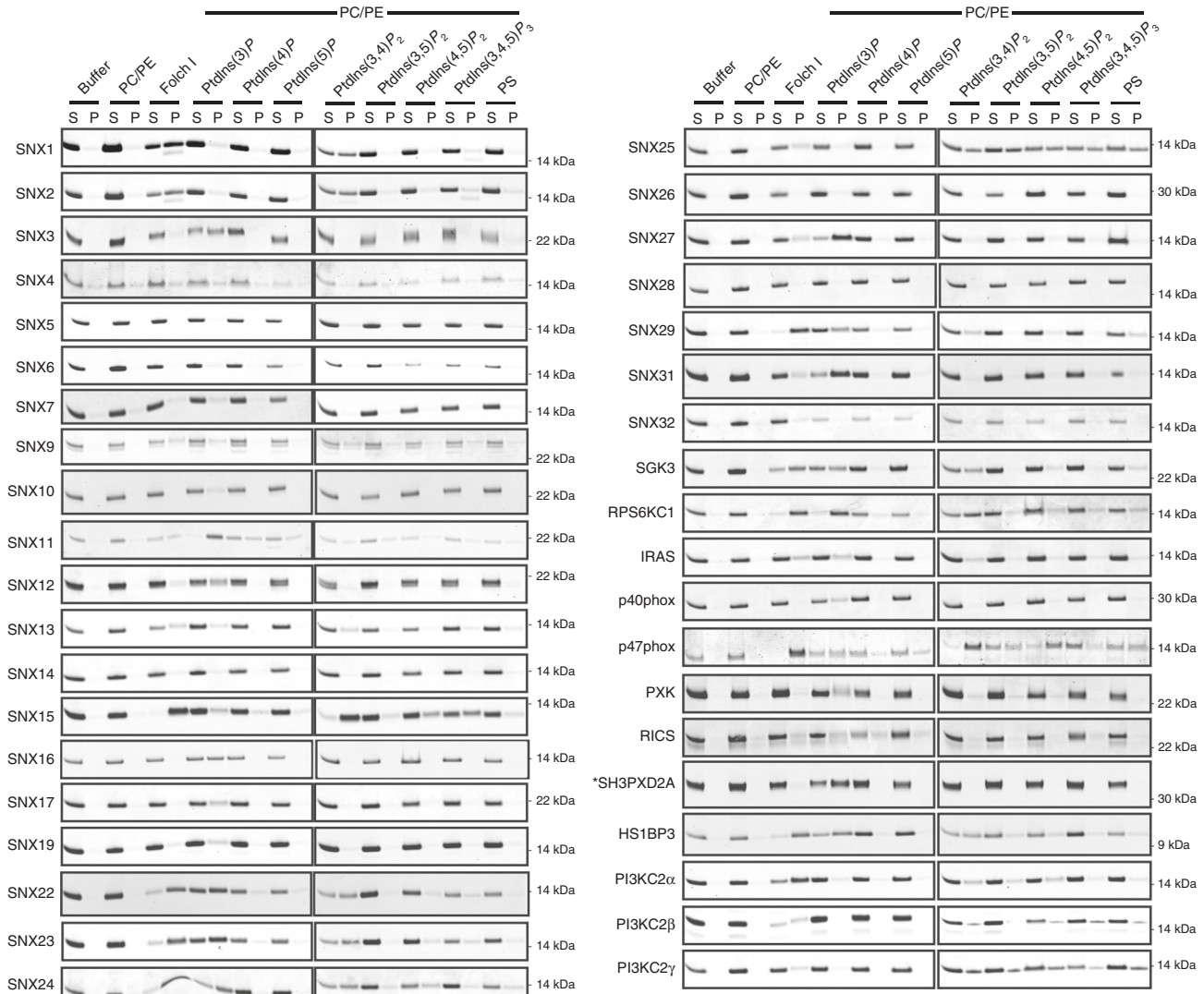

**Fig. 1** Membrane binding of the human PX domains by liposome pelleting assay. Bacterially expressed and purified PX domains with GST tags removed were incubated with artificial liposomes as indicated. Liposomes included POPC/POPE (PC/PE) as a negative control, Folch I as an indicator of broad membrane-binding activity, or PC/PE liposomes containing the specific lipids PtdIns3P, PtdIns4P, PtdIns5P, PtdIns(3,4)$P_2$, PtdIns(3,5)$P_2$, PtdIns(4,5)$P_2$, PtdIns(3,4,5)$P_3$, or PS. Samples were subjected to ultracentrifugation followed by SDS-PAGE and Coomassie staining of the unbound supernatant (S) and bound pellet (P) fractions. Gel filtration profiles of the purified proteins are shown in Supplementary Figure 1. SH3PXD2A (indicated by *) is the only protein that has its GST-tag intact

activities. Folch membranes contain significant amounts of PS, PtdIns(4,5)$P_2$, and some other phosphoinositides, but do not appear to contain PtdIns3P[20]. In all instances where we observe Folch binding by PX domains, we also observe interactions with various other phosphoinositide species. These can be broadly separated into two categories: those that bind to other phosphoinositides and PtdIns3P and those that bind other phosphoinositides but not PtdIns3P.

We next used isothermal titration calorimetry (ITC) to measure the binding of selected PX domains to soluble phosphoinositide headgroup analogues (Supplementary Figures 2, 3; Supplementary Table 4). These experiments quantitatively supported the results of the liposome pelleting assays for all of the PX domains tested. As an orthogonal method of quantitation, we used biolayer interferometry (BLItz) to measure the interactions of PX proteins with different phosphoinositide-containing membranes (Supplementary Figures 4, 5; Supplementary Table 5). In these experiments, liposomes containing a small amount of biotinylated lipids were coupled to streptavidin

sensors, and dipped into solutions containing the PX domains at a single concentration of 20 μM. Again, the results of these liposome-binding experiments using BLItz correlated well with both ITC and liposome pelleting assays. The dissociation constants derived from BLItz analyses were generally in agreement with those from ITC and correlated well with the qualitative interactions observed in pelleting assays.

**Sequence and structural analyses of the PX domain family.** Overall our systematic binding experiments show that the human PX domains can be divided into four different groups based on their phosphoinositide-binding preferences: those that do not bind any lipids tested (Group I), those that bind the canonical lipid PtdIns3P (Group II), those that bind to other phosphoinositides but not PtdIns3P (Group III), and those that bind other phosphoinositides and PtdIns3P (IV) (Fig. 2a, b). We performed a phylogenetic analysis of the human PX domains with sequence alignments guided by secondary structure predictions and found

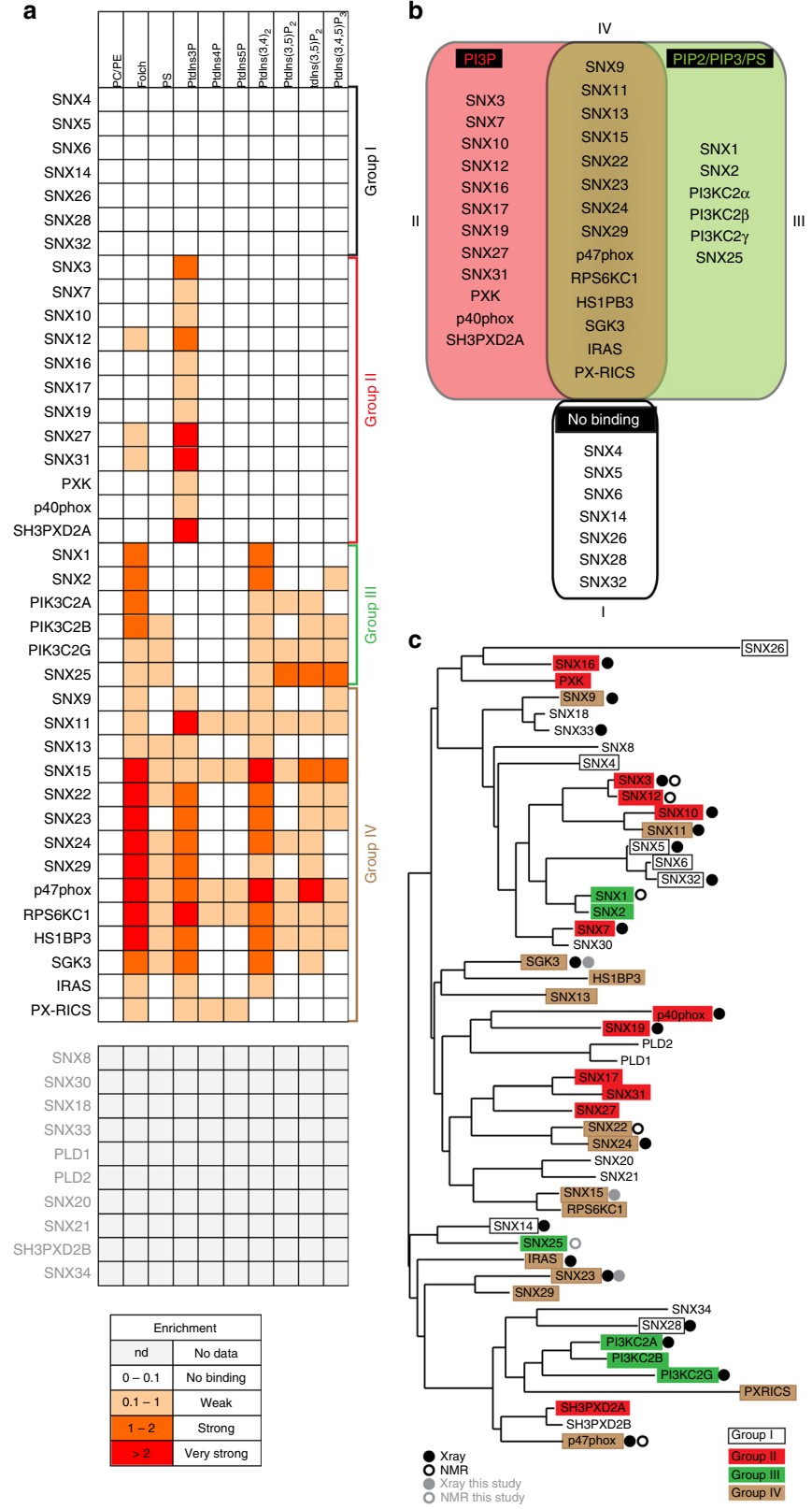

that different lipid-binding preferences were distributed quite broadly across different PX domain sub-families (Fig. 2c).

There are now many X-ray crystal and NMR structures deposited for the various members of the PX domain family (Figs. 2c, 3; Supplementary Table 1). We have also obtained two crystal structures of SGK3 and SNX23 (for which previous structures

exist), as well as a crystal structure of SNX15, the crystal structure of SNX32 (in complex with a peptide from the bacterial IncE peptide) and the NMR structure of SNX25 (Fig. 3; Supplementary Figures 6–8; Tables 1, 2). While the structures of SGK3, SNX23, and SNX25 all possess the canonical PX domain fold, our SNX15 construct appears to form a domain-swapped dimer both in the

**Fig. 2** Classification of the human PX domains based on their phosphoinositide-binding preferences. **a** The relative binding of each PX domain to phosphoinositide membranes was estimated based on the Coomassie-stained band intensities in Fig. 1. The enrichment or binding strength is defined as the ratio of the band intensity observed in the pellet (P) compared to the supernatant (S) fractions (P/S ratio). PX proteins that we could not purify (i.e., no data) are shown in white. **b** We have classified the PX domains into four groups. Group I proteins do not bind to any lipids under the conditions tested. Group II includes those that specifically bind to PtdIns3$P$. Group III members bind to PIP2 (PtdIns(3,4)$P_2$/PtdIns(3,5)$P_2$/PtdIns(4,5)$P_2$) and/or PIP3 (PtdIns (3,4,5)$P_3$) and/or PS, but not PtdIns3$P$. Group IV includes the PX proteins that bind to both PtdIns3$P$ and PIP2/PIP3/PS. **c** Phylogenetic analysis of the human PX domains performed with Phyology.fr[50], using sequence alignments guided by secondary structure predictions defined by PRALINE[51]. Proteins are color-coded based on their phosphoinositide-binding grouping, and those with known X-ray crystal or NMR structures are indicated

crystals and in solution (using analytical gel filtration chromatography) (Supplementary Figure 6A, B). It is unclear if this dimeric structure represents the normal state of the protein in the cell; however, the dimer does bind robustly to membranes as described above. SNX32 was crystallized in the presence of a peptide from the IncE protein of *Chlamydia trachomatis*, shown to bind SNX32 in previous studies[21]. Like SNX5, SNX32 possesses a highly extended helix-turn-helix structure between the PPK loop and α2, and the IncE peptide stabilizes the structure by forming an extended β-hairpin structure that binds to a highly conserved groove at the base of the helical extension (Supplementary Figure 7).

We next manually examined all of the known PX domain structures and compared these to structure-guided sequence alignments of the proteins (Figs. 3, 4). The PX domains are ~ 110–120 residues in length, and the core fold consists of three antiparallel β-strands followed by three α-helices[8,22]. Between the first and second α-helices, there is an extended stretch referred to as the PPK loop as it contains a relatively highly conserved ΨPxxPxK sequence (Ψ = large hydrophobic amino acids V, I, L, and M). Structures of p40phox, SNX9, and yeast Snx3p/Grd19p in complex with the PtdIns3$P$ headgroup reveal four critical residues that are required for canonical coordination with PtdIns3$P$. These are a sequential Arg and Tyr pair found at the junction between the β3 strand and the α1 helix, the Lys within the ΨPxxPxK sequence in the PPK loop, and an Arg within helix α2. As predicted, all of the proteins that bind to PtdIns3$P$ (groups II and IV) possess these essential four side-chains (Figs. 3, 4; Supplementary Figure 9). In contrast, all of the proteins that do not bind PtdIns3$P$ (groups I and III) lack at least one of these key side-chains. These sequence and structural considerations provide a clear logic for stereospecific canonical PtdIns3$P$ engagement by the core binding pocket.

Although the mechanism of binding to PtdIns3$P$ is thus well defined by the presence of these four strictly required side-chains, a large number of PX domains display additional or alternative lipid affinities without any known mode of interaction (groups III and IV). A careful analysis of the sequences and structures of the proteins in these groups show that a common feature is the presence of a His/Tyr side chain in the α1 helix, which is frequently found in conjunction with nearby basic Lys and Arg side-chains forming a region of positive electrostatic potential (Figs. 3, 4; Supplementary Figure 9). The basic side-chains are not precisely conserved with respect to their positions within the PX domain sequence but are in structurally adjacent locations at the C-terminus of the α1 helix or at the beginning of the PPK loop. Previous work suggested that this region within p47phox can mediate secondary lipid interactions[18], but whether these residues are involved in membrane binding in the PX family more generally is unknown.

**Noncanonical phosphoinositide-binding by SNX25.** To begin to assess if the His/Tyr-containing basic surface is involved in noncanonical lipid interactions, we took advantage of our SNX25 NMR structure and backbone amide assignments and performed HSQC titrations of different soluble phosphophoinositide headgroup analogues to map their binding sites (Fig. 5a; Supplementary Figure 10). In accord with liposome, ITC and BLItz experiments, SNX25 showed no interaction with PtdIns3$P$ or PtdIns4$P$ by NMR, but significant chemical shift perturbations were observed with di- and tri-phosphorylated phosphoinositides (PIP2 and PIP3)—PtdIns(3,4)$P_2$, PtdIns(3,5) $P_2$, PtdIns(4,5)$P_2$, and PtdIns(3,4,5)$P_3$. Although major perturbations were not seen in the His722 amide group, the amide of the nearby Lys734 at the start of the PPK loop consistently showed the largest chemical shift change. We mutated the Lys734 residue and found that this abolished binding to Folch liposomes (Fig. 5b). In addition, mutation of the nearby Lys762 also blocked membrane interaction. Notably, mutation of the Arg714 side chain present in the "canonical" pocket has no effect on membrane binding.

The affinity of the human SNX25 PX domain for di- and tri-phosphorylated phosphoinositides suggests it may associate with membranes other than PtdIns3$P$-enriched endosomes. To test this, we used *Saccharomyces cerevisiae* as a model cell system for membrane recruitment and expressed GFP-tagged versions of the SNX25 PX domain as well as the related proteins SNX13, SNX14, and SNX19, which all belong to the sub-family of SNXs possessing a regulator of G-protein signalling (RGS) domain. Previously we showed that SNX13 and SNX19 both have a preference for PtdIns3$P$ in vitro[20], and confirmed this again here (Fig. 1). In yeast cells, GFP-SNX19 PX domain shows clear localization to the PtdIns3$P$-enriched vacuole membrane as predicted; however, the SNX13 PX domain is predominantly cytoplasmic most likely due to a lower PtdIns3$P$ affinity ($K_d$)[20] (Fig. 6a). The GFP-SNX14 PX domain is also mostly cytoplasmic, which is in line with its complete lack of membrane binding in vitro[20]. In these experiments, the GFP-SNX25 PX domain was also cytoplasmic. To determine if this was simply due to a relatively modest phosphoinositide-binding affinity, we generated a construct with tandem SNX25 PX domains (residues 668–770) fused by a short linker (GGSSGG) to increase the binding avidity of the GFP-tagged protein. Interestingly, the GFP-SNX25$^{PX-PX}$ construct shows robust recruitment to the yeast plasma membrane, with no detectable localization to internal organelles (Fig. 6b). The major phosphoinositide at the plasma membrane is PtdIns(4,5)$P_2$, which in yeast is generated by the phosphorylation of PtdIns4$P$ by the sole PtdIns4$P$ 5-kinase Mss4p. To examine if GFP-SNX25$^{PX-PX}$ recruitment was PtdIns(4,5)$P_2$-dependent, we employed the *mss4*$^{ts}$ strain, which expresses a temperature-sensitive allele so that Mss4p is inactivated by temperature shift from 25 °C to 37 °C[23]. As shown in Fig. 6c in these cells, GFP-SNX25$^{PX-PX}$ is present at the plasma membrane at 25 °C but after shifting to 37 °C for 45 min is lost from the cell surface, confirming the 5-phosphoinositide-dependent membrane interaction in cells. Finally, to establish that this association is via the noncanonical phosphoinositide-binding site, we expressed the K734A mutant GFP-SNX25$^{PX-PX}$ construct that is deficient in membrane interaction in vitro (Fig. 6b). As predicted, this mutant was entirely cytoplasmic.

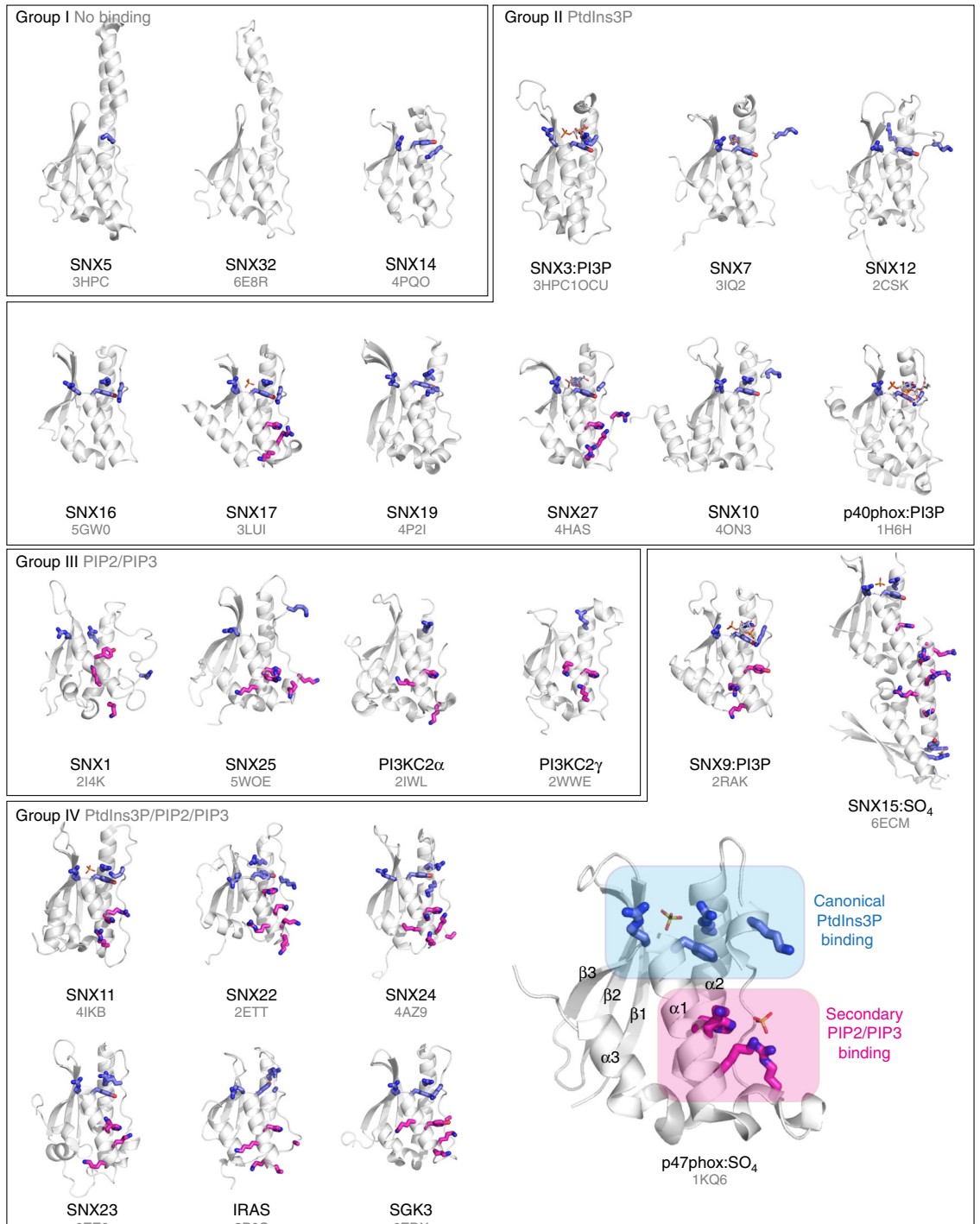

**Fig. 3** Known crystal and NMR structures of the PX domains. PX domains with known X-ray crystal or NMR structures are all shown in the orientation using a white cartoon representation. They are grouped according to their phosphoinositide-binding preferences (Fig. 2a). Known ligands including PtdIns3P and SO$_4{}^{2-}$ ions are shown in stick representation. Side-chains that are required for coordinating PtdIns3P in that canonical binding site are shown in blue with thick bonds. Shown in magenta are conserved His or Tyr side chain present in the α1 helix, and adjacent basic residues that together form a putative secondary binding site for alternative phosphoinositides. A sequence alignment highlighting these residues is shown in Fig. 4

**A general mode of noncanonical phosphoinositide-binding.**
Does our improved understanding of the canonical and secondary phosphoinositide-binding sites now allow us to predict the binding of other PX proteins? The detailed studies of SNX25 confirmed the importance of its secondary site for noncanonical phosphoinositide interactions. To extend this to the more general case, we performed mutagenesis of both the canonical and predicted secondary binding sites in other selected PX domains

(SNX1, SNX15, SNX23, SNX24, SNX25, and p47phox) and measured their association with membrane lipids using liposome pelleting assays and ITC (Fig. 7a; Supplementary Figure 11). To perturb the canonical binding site, we mutated either the first critical Arg residue that binds the 3-phosphate or the third Lys residue that binds the 1-phosphate. In the secondary site, we mutated the His/Tyr which is conserved in all the members of subgroups III and IV, alone or in combination with Lys and Arg

**Table 1 Summary of crystallographic structure determination statistics**

| | SNX32 PX domain-IncE | SGK3 PX domain | SNX23 PX domain | SNX15 PX domain (form 1) | SNX15 PX domain (form 2) |
|---|---|---|---|---|---|
| PDB ID | 6E8R | 6EDX | 6EE0 | 6ECM | 6MBI |
| *Data collection* | | | | | |
| Space group | I121 | I4 | P12$_1$1 | P4$_3$2$_1$2 | P4$_3$2$_1$2 |
| Cell dimensions | | | | | |
| $a, b, c$ (Å) | 90.8, 46.0, 134.2 | 74.3, 74.3, 52.0 | 46.8, 92.8, 47.0 | 54.5, 54.5, 83.6 | 60.3, 60.3, 85.2 |
| $\alpha, \beta, \gamma$ (°) | 90, 105.6, 90 | 90.0, 90.0, 90.0 | 90.0, 91.3, 90.0 | 90.0, 90,0, 90.0 | 90.0, 90,0, 90.0 |
| Resolution (Å) | 43.76-2.26 (2.34-2.27) | 42.59-2.009 (2.06-2.01) | 47.03-2.52 (2.62-2.52) | 45.65-2.35 (2.43-2.35) | 49.22-2.83 (2.98-2.83) |
| $R_{merge}$ | 0.151 (0.894) | 0.122 (0.181) | 0.062 (0.684) | 0.076 (1.074) | 0.094 (0.728) |
| $R_{pim}$ | 0.063 (0.540) | 0.035 (0.051) | 0.039 (0.428) | 0.015 (0.216) | 0.028 (0.210) |
| $<I>/\sigma I$ | 10.7 (2.5) | 13.9 (9.1) | 8.8 (1.3) | 20.9 (3.1) | 14.4 (2.6) |
| Total observations | 187434 (15896) | 126993 (9306) | 46543 (5123) | 147768 (12430) | 50092 (7141) |
| Unique observations | 25107 (2222) | 9533 (682) | 13562 (1488) | 5645 (505) | 4100 (565) |
| Completeness (%) | 99.7 (97) | 99.9 (99.3) | 99.0 (96.7) | 99.4 (94.3) | 99.4 (96.2) |
| Multiplicity | 7.5 (7.2) | 13.3 (13.6) | 3.4 (3.4) | 26.2 (24.6) | 12.2 (12.6) |
| Half-set correlation (CC(1/2)) | 0.996 (0.726) | 0.997 (0.990) | 0.998 (0.706) | 1.000 (0.982) | 0.999 (0.960) |
| *Refinement* | | | | | |
| $R_{work}/R_{free}$ | 0.186/0.238 | 0.187/0.213 | 0.175/0.247 | 0.244/0.299 | 0.280/0.314 |
| No. of atoms | | | | | |
| Protein | 2676 | 961 | 6584 | 834 | 860 |
| Solvent | 57 | 22 | 22 | 9 | 2 |
| Average $B$-factor (Å$^2$) | 26.3 | 27.1 | 80.7 | 79.0 | 103.7 |
| R.m.s deviations | | | | | |
| Bond lengths (Å) | 0.017 | 0.008 | 0.010 | 0.006 | 0.002 |
| Bond angles (°) | 1.312 | 0.980 | 1.520 | 1.036 | 0.536 |
| Ramachandran plot (%) | | | | | |
| Favored | 97.0 | 97.0 | 97.0 | 98.0 | 90.0 |
| Allowed | 2.7 | 3.0 | 2.2 | 1.0 | 9 |
| Outliers (%) | 0.3 | 0 | 0.8 | 1.0 | 1.0 |
| Clash score | 8 | 5 | 8 | 1 | 1 |

residues present at the C-terminus of helix α1 or N-terminus of the PPK loop.

Mutation of Lys213 in the canonical site of Group III protein SNX1 had no effect on the binding to Folch lipids or di- and tri-phosphorylated phosphoinositides (Fig. 7a; Supplementary Figure 11). In contrast, mutations Y194A/K196A/K200A in the secondary site of SNX1 resulted in complete loss of binding to Folch lipids and other phosphoinositides. Mutating the conserved His side chain in the Group III protein SNX25 also blocked membrane binding (Fig. 7a; Supplementary Figure 11), similar to the lysine mutations in the SNX25 secondary site described above (Fig. 5b). Mutation of the secondary site in the group IV proteins SNX23, SNX24, and p47phox also blocked binding to Folch lipids and various di- and tri-phosphorylated phosphoinositides, but these mutants still retained binding to PtdIns3P. The reciprocal mutation in the canonical binding site of p47phox (R44A) blocked binding to PtdIns3P, but retained binding to Folch and other lipids. The one outlier in these experiments was SNX15, which we believe is due to the unique nature of its domain-swapped structure. Although the protein sequence ostensibly places it in group III, and it does bind both PtdIns3P as well as di- and tri-phosphorylated phosphoinositides in our experiments, we find that mutation of the His and nearby Lys side-chains specifically ablates binding only to PtdIns(3,4)$P_2$.

Lastly, we confirmed the dual nature of the primary and secondary sites by competition experiments using ITC. For these experiments, the Group IV protein SGK3 was titrated with PtdIns3P or PtdIn(3,4)$P_2$, either alone, or following pre-incubation with the other lipid headgroup (Fig. 7b). In control experiments, pre-incubation with PtdIns3P in the cell blocks binding to PtdIns3P from the syringe as expected. Pre-incubation

of SGK3 with PtdIns3P in the cell, however, does not perturb binding to PtdIns(3,4)$P_2$ when titrated from the syringe, and pre-incubation with PtdIns(3,4)$P_2$ results in only a slight reduction in binding to PtdIns3P. This confirms that each lipid is able to bind to independent sites in the protein.

## Discussion

The synthesis and conversion of phosphoinositides from the abundant phosphatidylinositol precursor by phosphoinositide kinases and phosphatases is highly dynamic and leads to their specific accumulation in different compartments and within discrete membrane microdomains. Since its identification more than 20 years ago, the PX domain has been long established as a conserved structure for regulating interactions with membrane organelles of the secretory and endocytic system. PX domains are found in a wide variety of proteins with diverse functions, including cell signalling (e.g., PLD1 and 2, PIKC2α, β, and γ), vesicle trafficking (e.g., SNXs 1, 2, 3, 5, and 6), protein scaffolding (e.g., SH3PXD2A and B), and cytoskeletal transport (SNX23/KIF16B). Currently, 49 human proteins have been identified with the domain[6–8]. The commonly held view is that the domain primarily mediates interactions with the endosomal lipid PtdIns3P, promoting attachment to endosomal membranes in a manner that is also often dependent on coincident interactions with other lipids and proteins to increase binding avidity. However, PX domains also display unconventional and poorly understood phosphoinositide-binding specificities that are likely to be important for targeting the proteins to distinct subcellular compartments. In the current study, we have systematically analyzed the phosphoinositide-binding profile of more than

**Table 2 Structural Statistics of NMR Structure of the SNX25 PX domain[a]**

| Statistics | Value |
|---|---|
| *Experimental restraints*[b] | |
| Interproton distance restraints | |
| Total | 3024 |
| Intraresidue, $i = j$ | 716 |
| Sequential, $|i-j| = 1$ | 762 |
| Medium range, $1 < |i-j| < 5$ | 559 |
| Long range, $|i-j| \geq 5$ | 987 |
| Dihedral-angle restraints | |
| $\phi$ dihedral-angle restraints | 110 |
| $\psi$ dihedral-angle restraints | 105 |
| $\chi^a$ angle restraints | 47 |
| Total number of restraints per residue | 26.3 |
| *RMSD from mean coordinate structure, Å* | |
| All backbone atoms | 0.50 ± 0.17 |
| All heavy atoms | 0.79 ± 0.16 |
| Backbone atoms (residues 673–684, 687–698, 707–734, 750–783) | 0.21 ± 0.03 |
| Heavy atoms (residues 673–684, 687–698, 707–734, 750–783) | 0.58 ± 0.05 |
| *Stereochemical quality*[c] | |
| Residues in most favored Ramachandran region, % | 95.6 ± 1.0 |
| Residues in disallowed Ramachandran regions, % | 0 ± 0 |
| Favored side chain rotamers, % | 72.0 ± 2.8 |
| Unfavorable side chain rotamers, % | 13.6 ± 2.6 |
| Clash score, all atoms[d] | 0 ± 0 |
| Overall MolProbity score | 1.6 ± 0.1 (90.7 percentile) |
| *PDB ID* | 5WOE |
| *BMRB ID* | 30321 |

[a]All statistics are given as mean ± SD
[b]Only structurally relevant restraints as defined by CYANA are included
[c]According to Molprobity (molprobity.biochem.duke.edu)
[d]Defined as the number of steric overlaps >0.4 Å per thousand atoms

three-quarters of the human PX domain proteins, both qualitatively and quantitatively to define their distinct membrane-binding preferences.

Our experimental data, coupled with sequence and structural analyses, identify four specific groups of PX domains. There is a clear pattern of phosphoinositide-binding specificity related to the identity of key residues in either the canonical PtdIns3*P*-binding site or in the secondary binding site which is able to engage relatively nonspecifically with primarily di- and tri-phosphorylated phosphoinositides. Consistent with previously published work, we have shown that a significant fraction (12 of 39 tested) of the human PX domains bind exclusively to the canonical PtdIns3*P*, and this requires the strict conservation of four key side-chains that coordinate this headgroup with stereospecificity. Perhaps surprisingly, there are also a significant number of PX domains that show no ability to bind membranes in any of the conditions we have tested. With due consideration of the structural requirements for PtdIns3*P* engagement, however, this is clearly because they lack the necessary side chain determinants in the core binding sites.

The remaining PX domains all bind to a broader range of di- and tri-phosphorylated phosphoinositides. These interactions are shown to arise from the presence of a secondary binding site, centered on a semi-conserved His or Tyr side chain in the α1 helix, associated with nearby positively charged basic side-chains. Interestingly, many of these PX domains show a substantial preference for PtdIns(3,4)$P_2$ (e.g., SNX1, SNX2, SNX22, SGK3,

and p47phox), although the functional role of this is not yet clear from our work. The di- and tri-phosphorylated phosphoinositide-binding domains can be further classified into those that can also bind PtdIns3*P* (Group IV) and those that cannot (Group III). When the secondary site is present with an intact canonical binding site then both PtdIns3*P* and other phosphoinositides can promote membrane association, which will likely contribute to overall membrane avidity. But if the canonical site is altered then only di- and tri-phosphorylated phosphoinositide-binding is seen. There are several exceptions that we observe in our studies. First, we find that although SNX11 shows a strong preference for PtdIns3*P* it still binds to a variety of other membranes, despite lacking the His/Tyr residue within the α1 helix (Fig. 1). Examination of its structure shows that it does still possess a large basic surface in this secondary region that we speculate may be mediating these interactions (Fig. 3). Second, the SNX17 and SNX27 PX domains show strong preference for PtdIns3*P* (Fig. 1), yet both possess the His-basic patch forming that appears to be a secondary binding site (Figs. 3, 4; Supplementary Figure 8). It is possible that this plays a weak allosteric role in membrane binding and orientation of the PX domain, and we do detect weak binding of SNX27 to Folch liposomes supporting this notion (Fig. 1). Last, we observe an unusual domain-swapped structure and membrane-binding activity of the SNX15 PX domain. Interestingly, our binding data are similar to that shown previously for the isolated SNX15 PX domain; in contrast, however, the full-length SNX15 protein bound relatively strongly to PtdIns3*P*[24]. Further work will be needed to clarify the functional oligomeric state and membrane binding of this protein. Apart from the proteins, we successfully purified and screened, our data also allows us to make predictions about the membrane-binding preferences of the remaining PX domains (Supplementary Figure 9). For example, both SNX8 and SNX30 are predicted to belong to the PtdIns3*P*-specific group II based on conservation of the canonical site but not the secondary site, while in contrast the PLD1 and PLD2 proteins are predicted to belong to group III, which is supported by previous studies[25,26]. These structure and sequence-based predictions should be equally applicable to any PX domain from any species. Indeed we have found that the PX domain of snazarus, the sole Drosophila homologue of SNX13, SNX14, SNX19, and SNX25, shows specific binding to di- and tri-phosphorylated phosphoinositides that may allow it to interact with the plasma membrane in vivo (W.M. Henne, unpublished).

There are several discrepancies between our binding data and previously published studies. In our work, the PX domains that show no membrane binding include the closely related SNX5, SNX6, and SNX32. SNX5 was previously suggested by NMR to bind weakly to the plasma membrane-enriched PtdIns(4,5)$P_2$[27], while dot-blots and liposome-binding experiments with the SNX6 PX domain suggested an interaction with PtdIns3*P* and the trans-Golgi lipid PtdIns4*P*[28]. These discrepancies might be due to differences in experimental methodology, or because the affinities are too low to detect by our approaches. However, based both on our binding data and the lack of recognizable binding sites in the SNX5 and SNX32 structures (Fig. 3), we believe it is most likely that these PX domains in fact do not possess any specificity for phosphoinositides. It should be noted that regardless of whether their PX domains bind to lipids, the full-length SNX5, SNX6, and SNX32 proteins are clearly able to associate with endosomal membranes in vivo, which is likely to involve their C-terminal BAR domains and their formation of heterodimers with SNX1 or SNX2. Finally, although it does not preclude lipid binding, it is now firmly established that the SNX5, SNX6, and SNX32 PX domains are able to mediate protein–protein interactions[21,29,30] (Supplementary Figure 7), suggesting a divergent role for these proteins. With respect to SNX1 and the closely related SNX2, we

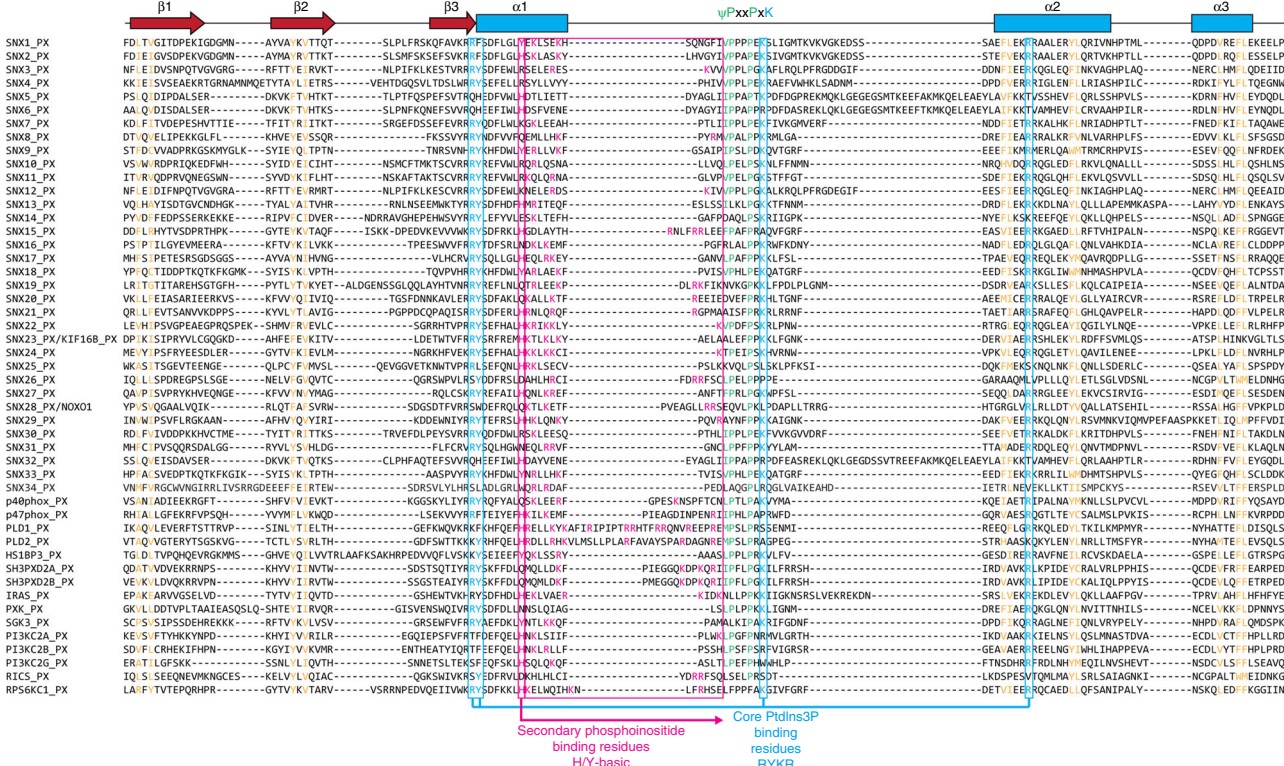

**Fig. 4** Secondary structure-guided sequence alignment of the human PX domains. Amino acid sequence alignment of the human PX domains guided by secondary structure predictions defined by PRALINE[51]. Where available, the precision of the sequence alignments was also manually curated by structural comparisons of the PX domains. As a reference, the secondary structure elements from the crystal structure of the PX domain of p40[phox] are indicated at the top. The four critical residues involved in canonical phosphoinositide (PtdIns3P) recognition are shown in blue. The noncanonical phosphoinositide-binding residues are highlighted in magenta. These include either a His or Tyr at the first position within helix α1, followed by a stretch of basic Lys/Arg residues leading to the polyproline loop containing the ΨPxxPxK sequence motif (Ψ = hydrophobic side chain)

observe binding only to Folch membranes and PtdIns(3,4)$P_2$, but not other lipids. This is in contrast with early studies that suggested binding of SNX1 to PtdIns3P and PtdIns(3,5)$P_2$[31]. We believe this difference is potentially due to the use of the full-length SNX1 protein in these previous experiments, with its additional membrane interacting BAR domain. A final difference we observe is with the PX domain PI3KC2α. Previous studies using surface plasmon resonance with membrane mimetics suggested a strong preference for PtdIns(4,5)$P_2$[19]. Although we do observe binding to PtdIns(4,5)$P_2$ both here and previously[32], we do not see the same restricted specificity, with other di- and tri-phosphorylated phosphoinositides binding with similar affinity. One final consideration with respect to PX domain membrane binding relates to their potential bilayer insertion. Although not addressed in this study, previous work has shown that hydrophobic residues adjacent to the canonical and secondary phosphoinositide-binding sites can also penetrate the lipid bilayer[19,26,33–35]. Thus membrane recruitment is likely to involve coupling of both specific headgroup interactions with hydrophobic lipid association.

In summary, our work helps to define the distinct phosphoinositide-binding classes of human PX domains based on their sequences and structures. By using strict criteria for protein quality, employing orthogonal methods to measure both membrane and headgroup binding, and by performing a systematic analysis of a large number of proteins in parallel, we have attempted to avoid issues of inconsistency that can arise from comparing data across multiple studies that often use different sample preparation methods and experimental approaches. Given the large range of cellular functions that are mediated the PX

domain proteins, we hope that the data presented here will provide a resource that enhances future studies of this important family of membrane-associated molecules.

## Methods

**Molecular biology and cloning.** The cDNA synthetic genes encoding the PX domains of human PX superfamily of proteins (Supplementary Table 2) optimized for *Eschericia coli* expression were synthesized by Genscript Corporation (USA), and subsequently cloned into the pGEX-4T-2 vector for expression as N-terminal GST fusion proteins with thrombin cleavage sites. The His/Tyr in the noncanonical phosphoinosidite-binding site along with the neighboring Lys/Arg were mutated to Ala using the QuikChange II site-directed mutagenesis protocol (Stratagene) and primers are listed in Supplementary Table 3.

**Recombinant protein expression and purification.** The plasmids encoding GST-fusion PX domains were transformed into BL21(DE3)/pLysS *E. coli* cells (Promega), and expressed in LB broth at 37 °C until $A_{600}$ reached 0.8. The cultures were induced with 0.5 mM isopropyl 1-thio-β-D-galactopyranoside and allowed to grow at 20 °C overnight, and cells were harvested by centrifugation (6000 × *g*, 10 min, 4 °C). The cell pellet was resuspended in lysis buffer (50 mM Tris (pH 8.0), 300 mM NaCl, 50 µg/ml of benzamidine, 100 units DNaseI, and 2 mM β-mercaptoethanol). The cells were lysed by mechanical disruption at 30 kpsi using a constant systems cell disrupter. The lysate was clarified by centrifugation at 50,000 × *g* for 30 min at 4 °C. Proteins were purified using affinity chromatography from the clarified lysate. The purification was performed on a glutathione–Sepharose (GE healthcare) gravity column, and the GST tag was cleaved with the addition of thrombin on to the beads with overnight incubation at room temperature. The proteins were eluted in 50 mM Tris (pH 8.0), 300 mM NaCl, and 2 mM DTT. The eluted affinity purified GST-cleaved proteins were finally subjected to size exclusion chromatography using a superdex-200 16/600 Hiload column, pre-equilibrated with 50 mM Tris (pH 8.0), 100 mM NaCl and 2 mM DTT, attached to an AKTA pure (GE Healthcare). The purified protein was concentrated to 10 mg/ml using a Centricon Ultra-3 kDa centrifugal filter (Millipore, USA) for crystallization, ITC, BLitz, and Liposome pelleting experiments. The concentration of protein was determined by the Biorad Protein Assay (Biorad, USA) after each purification step. For the

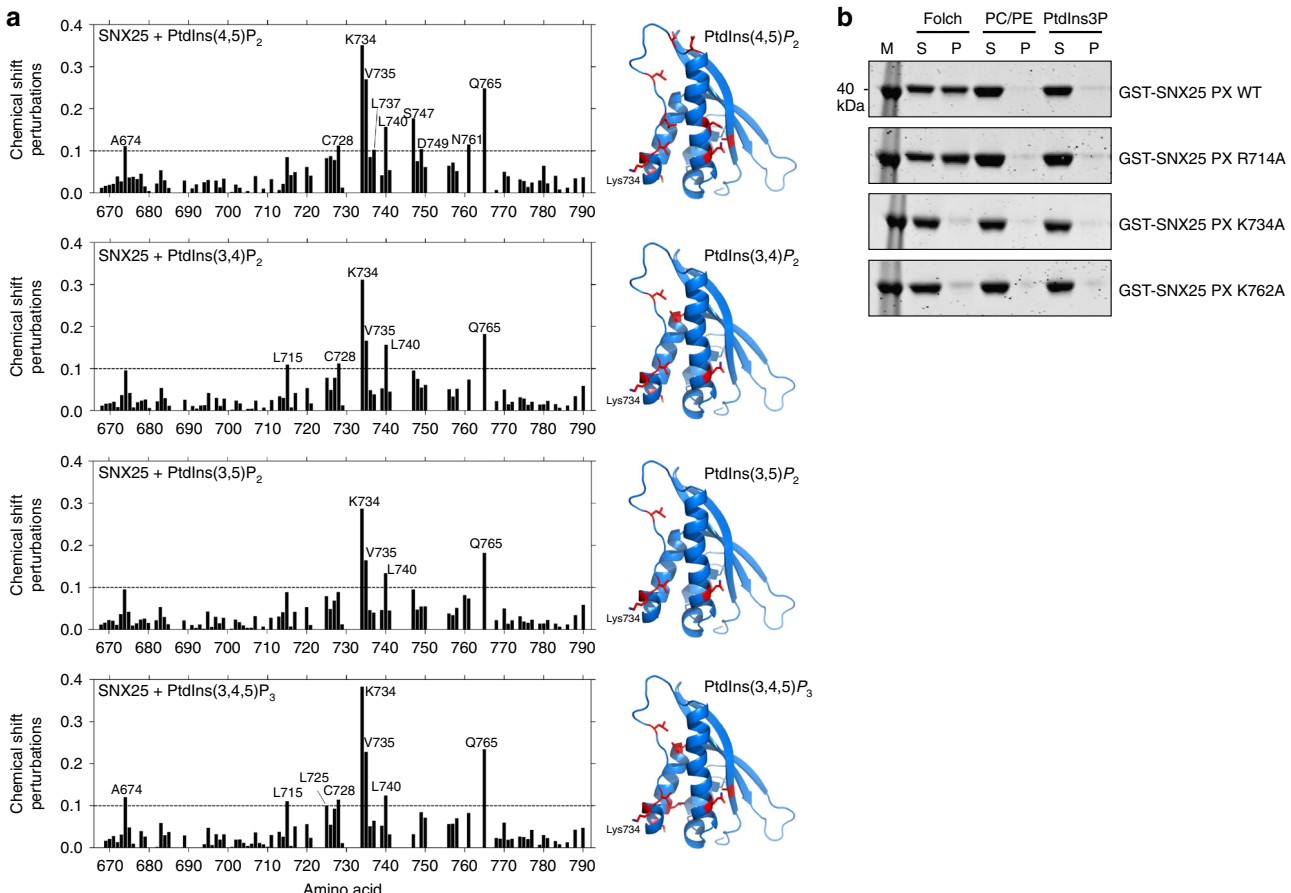

**Fig. 5** Binding of SNX25 to phosphoinositides by NMR. **a** Plot of the average chemical shift changes in the 2D $^1H$-$^{15}N$-HSQC spectra of the SNX25 PX domain upon addition of the indicated phosphoinositides. The SNX25 PX domain is shown in ribbon representation, with residues showing the greatest chemical shift changes The residues displaying significant chemical-shift changes on addition of the indicated phosphoinositides colored in red ($\Delta\delta > 0.1$ ppm; $\Delta\delta = [(0.17\Delta NH)^2 + (\Delta HN)^2]^{1/2})$[45]. **b** Liposome pelleting assay for GST-SNX25 PX domain and mutants. GST-tagged PX domains were incubated with the indicated artificial liposomes, subject to ultracentrifugation, and the pellet 'P' and supernatant 'S' fractions analyzed by SDS-PAGE and Coomassie staining

GST-tagged SNX13, SNX14, SNX19, and SNX25 proteins, the same procedure was used, but the thrombin cleavage step was omitted, and the proteins were eluted from glutathione–Sepharose with 20 mM glutathione in the elution buffer.

**Crystal structure determination**. The PX domain of SNX15, SNX23, and SGK3 were gel-filtered into buffer containing 50 mM Tris (pH 8.0), 100 mM NaCl, 2 mM DTT, and concentrated to 10 mg/ml for crystallization at 20 °C. The protein was supplemented with 10 mM DTT before setting up hanging-drop crystallization screens using a mosquito liquid handling robot (TTP LabTech). SNX15 PX domain from one crystal was crystallized in 100 mM Tris (pH 8.5), 200 mM LiSO₄, 24% PEG4000 and 10% glycerol. SNX15 PX domain from 2 crystals were crystallized in 4 M NaCl, 0.1 Tris (pH 8.5), with the addition of 30 mM inositolhexakisphosphate (IP6). Despite the presence of IP6, no bound ligand was observed in the electron density. The SNX23 PX domain was crystallized in 50 mM Tris (pH 8.5), 200 mM NaCl, and 25% PEG3350. The SGK3 PX domain was crystallized in 100 mM sodium citrate and 2 M sodium malonate (pH 5.0). For crystallization of the SNX32 complex with IncE, the synthetic peptide used for protein crystallization was purchased from Genscript (USA) (IncE residues PANGPAVQFFKGKNG-SADQVILVTQ). For the crystallization set up, peptides were weighed and dissolved in water to make a stock peptide concentration of 10 mM. This was diluted down to 2x molar excess to the protein molar concentration. The SNX32 PX domain (12 mg/ml) was directly mixed with the IncE peptide at a 1:2 molar ratio of protein to peptide and incubated on ice for 1 h. Ninety-six-well crystallization hanging-drop screens using commercial kits were set up using a Mosquito Liquid Handling robot (TTP LabTech) at 20 °C in the UQ ROCX facility. These plates were incubated at 20 °C in a Rockimager storage hotel (Formulatrix). Optimized diffraction-quality crystals of SNX32-IncE were obtained using streak seeding in sitting drop vapor diffusion plates. The optimized crystallization condition was 20% PEG 8000, 0.1 M TRIS (pH 8.0), 0.01 M MgCl₂.

Data were collected at the Australian Synchrotron MX1 and MX2 Beamlines. iMOSFLM[36] was used to integrate the data, and AIMLESS[37] was used for data

scaling in the CCP4 suite[38]. SNX23, SGK3, and SNX32-IncE were solved by molecular replacement using PHASER[39]. The structure of SNX15 was solved using single wavelength anamolous dispersion (SAD), and the phases were calculated using the peak wavelength data of Selenium with AUTOSOL using the PHENIX suite[40]. The solution from AUTOSOL was built using AUTOBUILD and the resulting model was rebuilt with COOT[41] followed by repeated refinement runs and model building with PHENIX.refine and COOT. Structure images were prepared with PYMOL (pymol.org).

**Phospholipids**. POPC (1-palmitoyl-2oleoyl-sn-glycero-3-phosphocholine) (catalog no. 850475 P), POPE (1-palmitoyl-2oleoyl-sn-glycero-3-phosphoethanolamine) (catalog no. 850757 P), DOPS (1,2-dioleoyl-sn-glycero-3-phosphoserine) (catalog no. 850150 P), and biotinylated POPE (1-palmitoyl-2oleoyl-sn-glycero-3-phos-phoethanolamine-N-(biotinyl)) (catalog no. 870285 P) were purchased from Avanti Polar Lipids. PI (1,2-dioleoyl-sn-glycero-3-phospho-(1'-myo-inositol)) (catalog no. P-0016), PI(3)P (1,2-dioctanoyl-sn-glycero-3-(phosphoinositol-3-phosphate)) (catalog no. P-3016), PI(4)P (1,2-dioleoyl-sn-glycero-3-phospho-(1'-myo-inositol-4'-phosphate)) (catalog no. P-4016), PI(5)P (1,2-dioleoyl-sn-glycero-3-phospho-(1'-myo-inositol-5'-phosphate)) (catalog no. P-5016), PI(3,4)P₂ (1,2-dioctanoyl-sn-glycero-3-phospho-(1'-myo-inositol-3',4'-bisphosphate)) (catalog no. P-3416), PI(3,5)P₂ (1,2-dioctanoyl-sn-glycero-3-phospho-(1'-myo-inositol-3',5'-bisphosphate)) (catalog no. P-3516), PI(4,5)P₂ (1,2-dioctanoyl-sn-glycero-3-phospho-(1'-myo-inositol-4',5'-bisphosphate)) (catalog no. P-4516), and PI(3,4,5)P₃ (1,2-dioctanoyl-sn-glycero-3-phospho-(1'-myo-inositol-3',4',5'-trisphosphate)) (catalog no. P-3916) lipids were obtained from Avanti Polar Lipids, Inc.

**Liposome preparation**. All the phosphoinositides were protonated prior to usage. In brief, powdered lipids were resuspended in chloroform (CHCl₃) and dried under argon. Dried lipids were then left in a desiccator for 1 h to remove any remaining moisture. Dried lipids were resuspended in a mixture of CHCl₃:methanol (MeOH):1 N hydrochloric acid in a 2:1:0.01 molar ratio, lipids were dried once

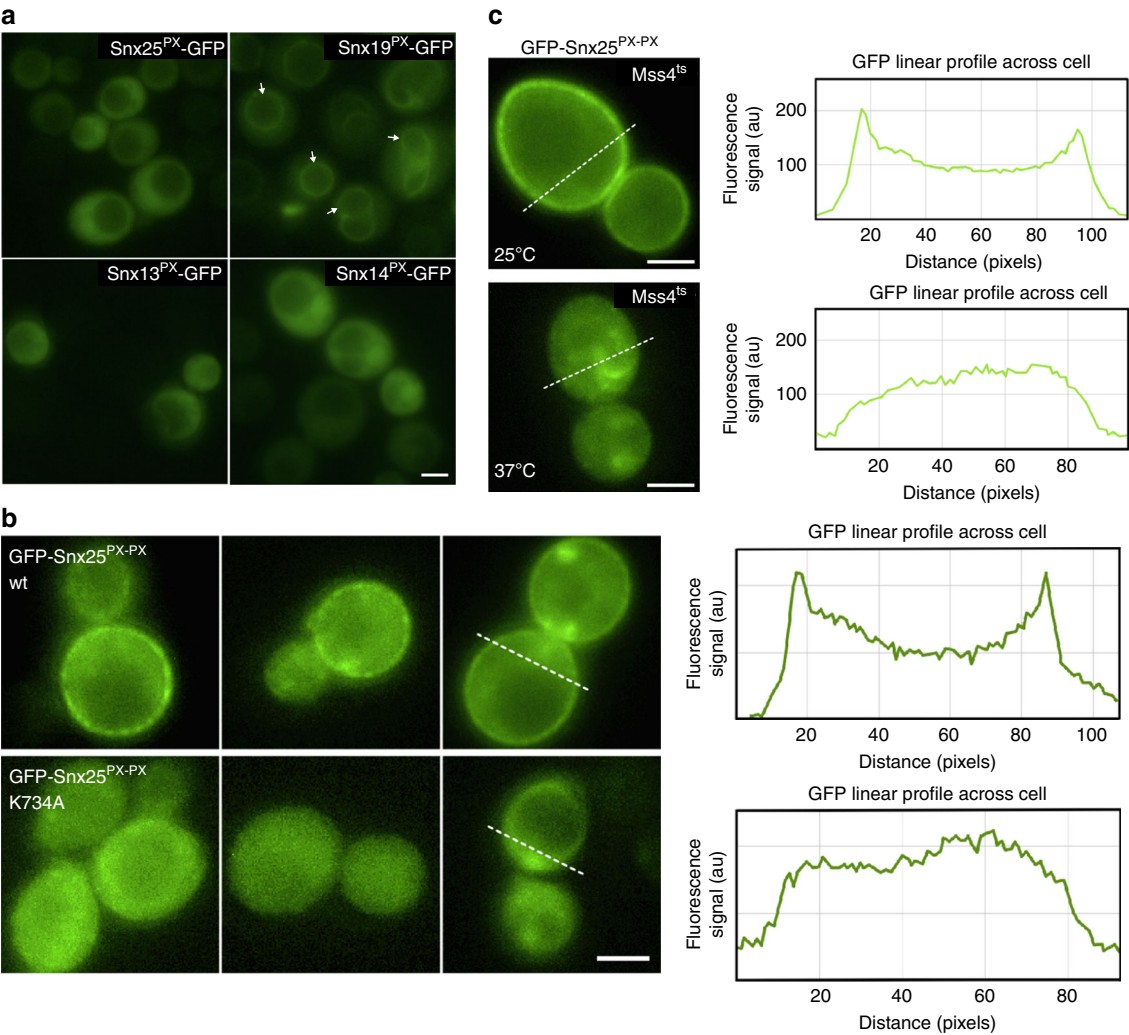

**Fig. 6** The SNX25 PX domain is recruited to the plasma membrane via PtdIns(4,5)$P_2$ interaction. **a** GFP-tagged PX domains from human SNX25, SNX13, SNX14, and SNX19 were expressed in *S. cerevisiae* and their localization examined by confocal fluorescence microscopy. Using single PX domains, only SNX19 shows membrane recruitment to the PtdIns3P-enriched vacuole. White arrows indicate examples of the vacuolar recruitment of the SNX19 PX domain. **b** The tandem GFP-SNX25$^{PX-PX}$ construct is recruited to the plasma membrane in yeast. This localization is blocked by the K734A lipid-binding mutation of the tandem PX domains. GFP linear profiles were made using RGB Profiler (ImageJ), and span the yeast cell. **c** In yeast with the temperature sensitive *mss4$^{ts}$* allele, plasma membrane recruitment of GFP-SNX25$^{PX-PX}$ is normal at the permissive temperature of 25 °C, but lost at the restrictive temperature of 37 °C. This demonstrates that membrane interactions requires the synthesis of PtdIns(4,5)$P_2$ by the sole yeast PI4P 5-kinase Mss4p

again and allowed to desiccate. Lipids were then resuspended in CHCl$_3$:MeOH in a 3:1 ratio dried once again under argon. Finally, dried lipids were resuspended in CHCl$_3$ and stored at −20 °C.

Lipid stock solutions were mixed to the desired molar ratios and dried under argon. To prepare control liposomes, POPC and POPE were mixed in a 90:10 molar ratio, for BLiTz experiments, liposomes were doped with 0.5% biotinylated POPE. Liposomes containing phosphoinositides were prepared by mixing POPC, POPE, and PIPs in a 80:10:10 molar ratio, respectively. Thirty percent POPS was used for POPC:POPE:POPS. Dried lipids were hydrated in 25 mM HEPES (pH 7.2), and 220 mM sucrose to obtain a suspension of multilamellar liposomes containing sucrose. This solution was then freeze-thawed five times to produce unilamellar liposomes. Liposomes were diluted 1:5 in 25 mM HEPES (pH 7.2), and 125 mM NaCl solution. The solution was then centrifuged at 250,000 *g* to remove sucrose from the medium and maintain osmolarity. The pelleted liposomes were resuspended in 25 mM HEPES (pH 7.2), and 125 mM NaCl solution to the desired concentration of 0.5 mM. All liposomes were used within 1 day of preparation.

**Liposome pelleting**. In total, 20 μM of the protein of interest was added to a final volume of 100 μl of the liposome solution. This solution was left at room temperature for 25 min to allow for protein–liposome interaction. After incubation, the solution was centrifuged at 400,000 *g* for 30 min. Supernatant and pellet fractions were separated and the pellet was resuspended in 100 μl of 25 mM HEPES (pH 7.2), and 125 mM NaCl, samples were then collected for analysis on a precast 4–12%

bis-tris gel (Novex) by coomassie staining. The binding of the PX proteins–phosphoinositides interactions within the SDS-PAGE has been further quantified by measuring the protein band intensities with ImageJ[42]. We have defined the binding as very strong if there is an enrichment of the fraction of Pellet/Supernatant (P/S) > twofold, strong binding if the fraction of pellet/supernatant (P/S) = 1–2-fold, weak binding if the fraction of pellet/supernatant (P/S) = 0.1–1-fold, and no binding if the fraction of pellet/supernatant (P/S) = 0–0.1-fold.

**Isothermal titration calorimetry (ITC)**. The affinities of human PX proteins and various mutants for PtdIns*P*s were determined using a Microcal iTC200 instrument (Malvern, UK). Soluble diC8 PtdIns*P*s were purchased from Echelon Biosciences. Experiments were performed in 50 mM Tris-HCl pH 8.0, 100 mM NaCl. The PtdIns*P*s at 0.5 mM were titrated into 0.020 mM proteins in 13 × 3.22 μl aliquots at 25 °C. The dissociation constants ($K_d$), enthalpy of binding ($\Delta H$), and stoichiometries (N) were obtained after fitting the integrated and normalized data to a single-site binding model. The stoichiometry was refined initially, and the value obtained was close to 1; then, N was set to 1.0 for calculation. The apparent binding free energy ($\Delta G$) and entropy ($\Delta S$) were calculated from the relationships $\Delta G = RT\ln(K_d)$ and $\Delta G = \Delta H − T\Delta S$. All experiments were performed at least in duplicate to check for reproducibility of the data.

**Biophysical interaction using biolayer interferometry**. Protein–lipid interactions were determined using the biolayer interferometry from the BLItz system (Pall

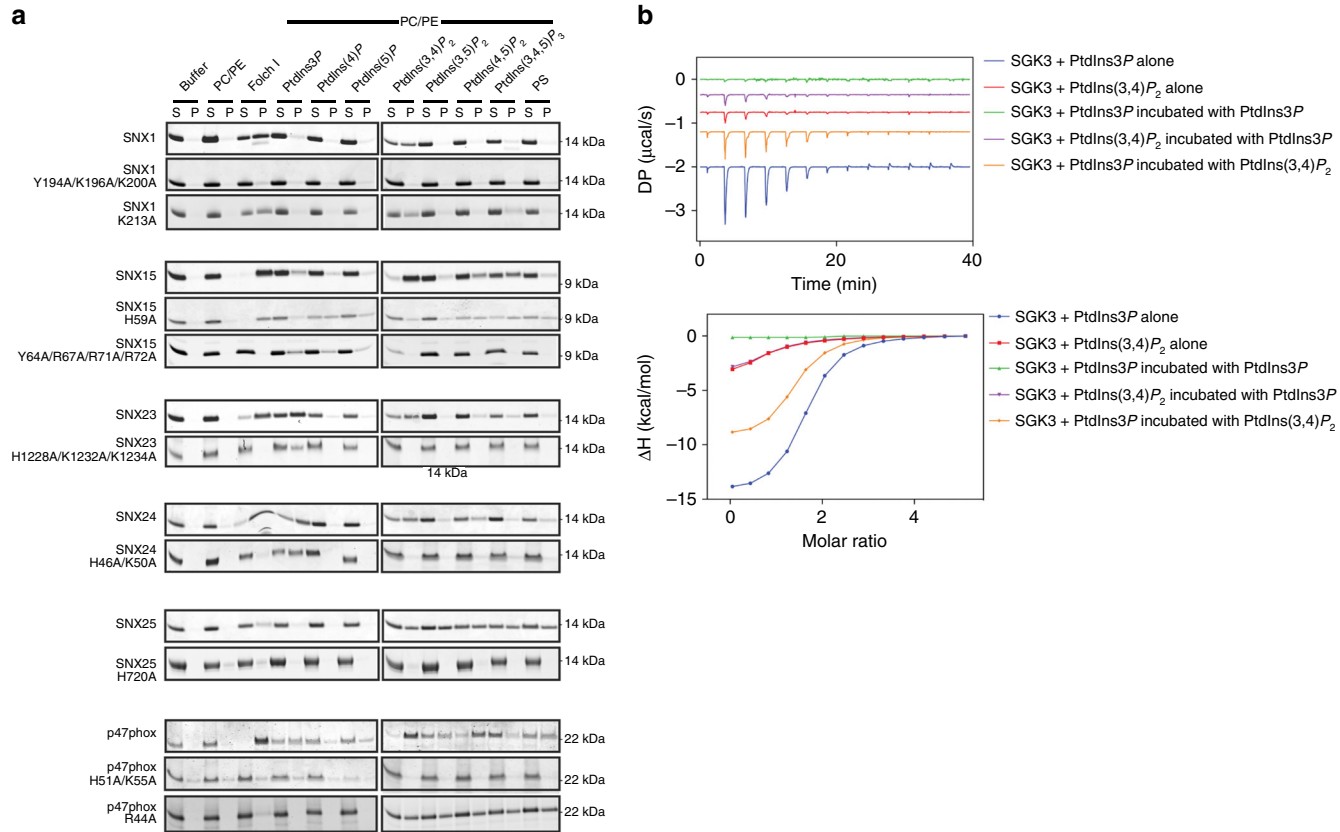

**Fig. 7** Mutagenesis of canonical and noncanonical sites in PX domain proteins. **a** Selected bacterially expressed and purified PX domains and their mutants with GST tags removed were incubated with artificial liposomes as indicated. Liposomes included POPC/POPE (PC/PE) as a negative control, Folch I as an indicator of broad membrane-binding activity, or PC/PE liposomes containing the specific lipids PtdIns3P, PtdIns4P, PtdIns5P, PtdIns(3,4)$P_2$, PtdIns(3,5)$P_2$, PtdIns(4,5)$P_2$, PtdIns(3,4,5)$P_3$, or PS. Samples were subjected to ultracentrifugation followed by SDS-PAGE and Coomassie staining of the unbound supernatant (S) and bound pellet (P) fractions. **b** Water-soluble PtdIns3P and PtdIns(3,4)$P_2$ headgroup analogues (500 μM) were titrated into the SGK3 PX domain (20 μM) and measured by ITC. To test for competition between the two lipids, the SGK3 PX domain was pre-incubated with the indicated lipid before titration with the other. Top panels show the raw data and bottom panels represent the integrated and normalized data fit with a 1:1 binding model. The binding affinities ($K_d$) are provided in Supplementary Table 1

ForteBio, USA). Protein–lipid interactions were observed by immobilizing 500 μM of biotinylated liposomes on a streptavidin biosensor. After immobilization, the sensor was washed with buffer containing 50 mM Tris (pH 8.0), 100 mM NaCl, and 0.1% BSA to prevent nonspecific association. In total, 25 μM of proteins were added to the sensor and the change in binding (nm) was measured. Proteins were then allowed to disassociate from the probe in the buffer previously mentioned. The data were processed and plotted using the Prism Software.

**Analytical gel filtration analysis of SNX15 PX domain**. The PX domain of SNX15 was subjected to analytical gel filtration on a Superdex 200 10/300 column (Amersham, GE healthcare) equilibrated with a buffer containing 50 mM Tris (pH 8.0), 100 mM NaCl, and 2 mM DTT. Overall, 100 -μl SNX15 PX domain (500 μM) was separated at 0.5 ml min$^{-1}$ at 4 °C, and compared with gel filtration standards curves. Protein elution was monitored by the absorbance at 280 nm. Molecular mass across the protein elution peak was calculated using the standard calibration curve of the markers.

**SNX25 NMR phosphoinositide titration experiments**. For NMR chemical shift titration studies, soluble phosphoinositide headgroup analogs with di-C8 aliphatic chains (Echelon Biosciences) were added to samples containing 50 μM of [15]N-labeled SNX25 PX domains in NMR buffer (20 mM HEPES (pH 7.0), 100 mM NaCl, 2 mM DTT, and 10% D$_2$O). Two-dimensional (2D) [1]H-[15]N HSQC spectra of the [15]N-labeled SNX25 PX domain was collected with increasing amounts of the soluble phosphoinositide headgroup analogs. All spectra were collected at 298 K on a Bruker 900-MHz spectrometer equipped with a cryoprobe and z-axis gradients. Spectra were processed using NMRPipe[43] and analyzed with the program CCPNMR[44]. Chemical shift changes were calculated as $\Delta\delta = [(0.17\Delta NH)^2 + (\Delta HN)^2]^{1/2}$[45].

**SNX25 NMR structure determination**. The structure of SNX25 PX domain was determined using heteronuclear NMR data acquired at 25 °C on a 900 MHz Bruker

Avance II + spectrometer equipped with a cryogenic probe. Uniformly [13]C/[15]N-labeled SNX25 PX domain at 0.8 mM in the NMR buffer was used. For backbone resonance assignments, 2D [1]H-[15]N-HSQC, 3D HNCACB, 3D CBCA(CO)NH, 3D HNCO, and 3D HBHA(CO)NH were acquired on the SNX25 sample using nonuniform sampling (NUS)[46] and processed using maximum entropy reconstruction[47]. Distance restraints were derived from 3D [13]C aliphatic/aromatic-edited [1]H,[1]H]-NOESY-HSQC and [15]N-edited [1]H,[1]H]-NOESY-HSQC spectra with a NOE mixing time of 120 ms. NOESY spectra were manually peak-picked and integrated using CCPNMR analysis 2.4.1[44]. The peak lists were then assigned and an ensemble of structures calculated automatically using the torsion angle dynamics package CYANA 3.97[48]. The tolerances used in the structure calculations were 0.03 ppm in both the direct and indirect [1]H dimension and 0.4 ppm for the [13]C and [15]N dimensions. Backbone and side chain dihedral-angle restraints (110 φ, 105 ψ, and 47 χ[1] angles) were derived from TALOS-N chemical shift analysis[49]; the restraint range was set to twice the estimated standard deviation. All X-Pro peptide bonds were clearly identified as *trans* on the basis of characteristic NOEs and the Cβ and Cγ chemical shifts for the Pro residues. CYANA was used to calculate 200 structures from random starting conformations, then 20 conformers with the lowest CYANA target function were chosen to represent the structural ensemble. During the automated NOESY assignment/structure calculation process, CYANA assigned 91.8% of all NOESY cross peaks (4774 out of 5199) for SNX25.

**Localization of PX domains in S. cerevisiae**. Living yeast were cultured in synthetic complete media containing 2% dextrose, and imaged while undergoing exponential growth using an EVOS FL fluorescent microscope (Thermo-Fisher). Human PX domain constructs were cloned into the pBP73G vector using BamHI/XhoI sites and expressed under the GPD promoter in SEY6210 yeast (genotype: MATα leu2-3,112 ura3-52 his3-Δ200 trp1-Δ901 suc2-Δ9 lys2-801; GAL). For the MSS4-ts experiments, AAY202 yeast (genotype: mss4Δ:HIS3MX6 YCplac111mss4ts-102 (LEU2 CEN6 mss4ts-102))[23] were grown at 25 °C into late exponential phase, then moved into a 37 °C shaker incubator for 45 min and

immediately imaged. Line scans were conducted using the RGB Profiler Plugin in ImageJ.

**Reporting Summary**. Further information on experimental design is available in the Nature Research Reporting Summary linked to this article.

## Data availability

Data supporting the findings of this manuscript are available from the corresponding author upon reasonable request. A reporting summary for this Article is available as a Supplementary Information file. Coordinates and structure factors for the PX domain of SNX23 and SGK3 have been deposited in the RCSB PDB with IDs 6EE0 and 6EDX, respectively. The SNX15 form 1 and form 2 structures are deposited with IDs 6ECM and 6MBI respectively. The SNX32 PX domain complex with IncE coordinates have PDB ID 6E8R. Raw diffraction data is available on the University of Queensland eSpace server. Coordinates for the SNX25 PX domain NMR structure have been deposited in the Protein Data Bank with accession number 5WOE. The chemical shift data has been deposited in the Biological Magnetic Resonance Bank under accession number 30321.

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

## Acknowledgements

The authors would like to acknowledge support from the staff and facilities of the University of Queensland Remote Operation Crystallization and X-ray (UQ ROCX) facility, and the Australian Synchrotron. This work is supported by funds from the Australian Research Council (ARC) (DP160101743) and National Health and Medical Research Council (APP1099114). MM is supported by an ARC Future Fellowship (FT10100925). BMC is supported by an NHMRC Senior Research Fellowship (APP1136021).

## Author contributions

M.C., Y.K.-Y.C., C.M., W.M.H., M.M., and B.C.: experimental design, data analysis, and paper preparation. R.F., B.P, S.D., K.-E.C, X.J., Z.Y., S.J.N., B.M., and A.B.: data analysis, paper preparation. R.D.T.: study concept, data analysis, and paper preparation.

## Additional information

**Competing interests:** The authors declare no competing interests.

