## [Peer Review File · Nature Communications]

Reviewers' Comments:

Reviewer #1:

Remarks to the Author:

In this study, Chandra and colleagues present a comprehensive analysis of PI-binding activities of almost the entire family of human PX domains. The authors evaluated selectivities of the PX domains for mono, di and tri phosphorylated PIs by liposome binding assays, measured Kds using ITC, investigated lipid-binding interfaces of some PX domains by NMR and mutagenesis, and determined several crystal and solution structures of the PX domains. The authors also provide a thorough comparative analysis of their results and previously published data.

The manuscript contains excellent quality data and nicely summarizes what is currently known about the PX domain family. It will be a valuable resource for many scientists working in the area of phosphoinositide signaling and membrane trafficking. I have only a few comments primarily on description of results that could be of help in further strengthening this manuscript.

1. Fig. 3: I would include PDB IDs in the figure legend.
2. It has been shown (Lee, JBC, 2006; Stahelin, JBC, 2003; Kutateladze, Prog Lipid Res, 2007 and refs therein) that some PX domains insert in membrane- this should be discussed.
3. Fig. 5: unclear as to why $0.17\delta\text{NH}$ was used in the equation. Also unclear what does a significant chem shift change represent.
4. Suppl Fig. S9: not sure whether measurements of Kds by NMR are essential here. The fast exchange regime and the binding curves show weak interactions, likely in the range of hundreds micromolar, cannot be a low micromolar range.

Reviewer #2:

Remarks to the Author:

The study by Chandra et al., is a tour de force study of phosphoinositide-binding PX domains. The manuscript reports the purification of 39 different PX domains and the characterization of these qualitatively by liposome interaction assays and quantitatively using ITC. Several structures of PX domains are reported (determined by X-ray crystallography and NMR). A second phosphoinositide binding site is characterized and the amino acid requirements elucidated. The authors divide PX domains into 4 groups depending on lipid binding preferences and their data allow them to predict the preferences of yet uncharacterized PX domains. I find that the work is interesting and generally well carried out. The data will surely be of great service to the membrane community in the years to come and I recommend publishing it after addressing the following points.

0) pg 4, lines 61-62: The authors state that previous studies of PX domain interaction with membranes have yielded conflicting results but do not list any references to back up this claim. Either include references or refer to your Table S1.

1) 39 PX domains were purified according to Table S2. Could the authors show SEC profiles or other data to give the reader a feeling for the quality of the preparations? Any issues with aggregation?

2) Table S1 reports the structure of the PX domain of SNX15 but the PDB code is not listed. Was this structure not deposited?

From Table S1: PX-MIT SNX15 XXXX X-ray/SO4 This study PI3P Liposome pelleting and PIP strip -

(77)

3) Table 1: Several of the X-ray diffraction data sets have been cut too conservatively with respect to resolution. In particular, pdb code 6EDX has an I/σ of 9 in the highest resolution shell and thus extends much further. It is a shame to have well diffracting crystals and then throw away good data that could be very useful in refinement. I suggest using CC 1/2 of around 70% as done for 6E8R and 6EE0 as a cut off.

Additionally, the authors should provide Ramachandran plot and rotamer outlier statistics for the structures shown in Table 1. The validation reports for the structures show that some of the PDBs have rather high percentage of rotamer and Ramachandran plot outliers. The authors should carefully examine these in a graphics program such as Coot and correct the structures (given the decent resolution this should really not be a problem).

4) Fig. 1: was care taken to display the pelleting assays using normalized settings? In some of the experiments the background appears very white and in some much darker. Are the different pelleting assays shown in Fig. 1 directly comparable?

5) For the ITC data shown in Table S3, could the authors please provide standard deviations for the N-values.

6) pg 10, lines 270-272: 'a certain number of PX domains do bind exclusively to the canonical PtdINS3P...'? Given all the data you present surely you can do better than that.

Reviewer #3:

Remarks to the Author:

The PX domain was first identified as PI3P-binding module almost two decades ago. Numerous biochemical and structural studies have shown since that PX domains can also interact with many other lipids. Now Chandra et. al. are reporting *tour de force* characterization of 39 of 49 human PX domains by three independent membrane binding analyses. They also performed structural analyses, including determination of a few PX domain structures, structural comparisons, and NMR titration of SNX25-PX-lipid binding, mutational studies, and subcellular localization analysis of selected PX domains in yeast cells. Overall, the work is technically solid and rigorous. It contains the comprehensive and well-organized information about lipid binding properties of PX domains, which would undoubtedly serve as a useful resource for those who study PX domain-containing proteins. The main concern is, however, its incremental nature. This report presents neither a technical breakthrough nor new structural and biological insight. Although the authors should be praised for employing such diverse and orthogonal methods to characterize lipid binding of PX domains, they all represent well established standard methods, hence lack of any new critical structural or functional information. Many previous studies have addressed the physiological significance of diverse lipid specificity of PX domains with in-depth biological studies. Besides classification of PX domains based on their lipid selectivity and some structural rationalization, however, this work does not add much to our understanding of biological insight into how PX domains mediate diverse cellular functions.

Revision of manuscript NCOMMS-18-33397

Editor comments

Your manuscript entitled "Classification of the human phox homology (PX) domains based on their phosphoinositide binding specificities" has now been seen by 3 referees. You will see from their comments below that while they find your work of interest, some important points are raised. We are interested in the possibility of publishing your study in Nature Communications, but would like to consider your response to these concerns in the form of a revised manuscript before we make a final decision on publication.

We therefore invite you to revise and resubmit your manuscript, taking into account the points raised. Please note that we do not share the concerns of Rev#3 that the manuscript is of incremental advance. Please highlight all changes in the manuscript text file.

Reviewer #1

In this study, Chandra and colleagues present a comprehensive analysis of PI-binding activities of almost the entire family of human PX domains. The authors evaluated selectivities of the PX domains for mono, di and tri phosphorylated PIs by liposome binding assays, measured Kds using ITC, investigated lipid-binding interfaces of some PX domains by NMR and mutagenesis, and determined several crystal and solution structures of the PX domains. The authors also provide a thorough comparative analysis of their results and previously published data.

The manuscript contains excellent quality data and nicely summarizes what is currently known about the PX domain family. It will be a valuable resource for many scientists working in the area of phosphoinositide signaling and membrane trafficking. I have only a few comments primarily on description of results that could be of help in further strengthening this manuscript.

1. Fig. 3: I would include PDB IDs in the figure legend.

For ease of reference the PDB IDs for the structures are now provided alongside the panels in Figure 3 itself.

2. It has been shown (Lee, JBC, 2006; Stahelin, JBC, 2003; Kutateladze, Prog Lipid Res, 2007 and refs therein) that some PX domains insert in membrane- this should be discussed.

A brief discussion of the ability of PX domains to insert into the membrane is now included on page 12 within the discussion section.

3. Fig. 5: unclear as to why $0.17\Delta\text{NH}$ was used in the equation. Also unclear what does a significant chem shift change represent.

We now cite the original reference for the equation for the normalised magnitude of NMR chemical shift calculations: "Localizing the NADP+ binding site on the MurB enzyme by NMR. Farmer et al., Nat. Struct. Mol. Biol. 1996".

4. Suppl Fig. S9: not sure whether measurements of Kds by NMR are essential here. The fast exchange

regime and the binding curves show weak interactions, likely in the range of hundreds micromolar, cannot be a low micromolar range.

We agree with the reviewer, and since these affinity estimates are not essential for the manuscript we have simply removed these panels from Supplemental Figure S9.

Reviewer #2

The study by Chandra et al., is a tour de force study of phosphoinositide-binding PX domains. The manuscript reports the purification of 39 different PX domains and the characterization of these qualitatively by liposome interaction assays and quantitatively using ITC. Several structures of PX domains are reported (determined by X-ray crystallography and NMR). A second phosphoinositide binding site is characterized and the amino acid requirements elucidated. The authors divide PX domains into 4 groups depending on lipid binding preferences and their data allow them to predict the preferences of yet uncharacterized PX domains. I find that the work is interesting and generally well carried out. The data will surely be of great service to the membrane community in the years to come and I recommend publishing it after addressing the following points.

0) pg 4, lines 61-62: The authors state that previous studies of PX domain interaction with membranes have yielded conflicting results but do not list any references to back up this claim. Either include references or refer to your Table S1.

*We thank the referee for this suggestion and now refer to **Table S1** (and references therein) in the text on Page 4.*

1) 39 PX domains were purified according to Table S2. Could the authors show SEC profiles or other data to give the reader a feeling for the quality of the preparations? Any issues with aggregation? *The proteins were generally well behaved, and indeed the monomeric nature of the proteins by gel filtration was assessed as a quality control prior to lipid binding experiments. The gel filtration profiles of the purified proteins are now provided as **Attachment 1**. We have not included these as supplementary figures in the revised manuscript but could do so if requested.*

2) Table S1 reports the structure of the PX domain of SNX15 but the PDB code is not listed. Was this structure not deposited? From Table S1: PX-MIT SNX15 XXXX X-ray/SO4 This study PI3P Liposome pelleting and PIP strip - (77)

*Apologies for the omission. This referred to our own SNX15 PX domain structure and the PDB ID has now been included in **Table S1**.*

3) Table 1: Several of the X-ray diffraction data sets have been cut too conservatively with respect to resolution. In particular, pdb code 6EDX has an I/σ of 9 in the highest resolution shell and thus extends much further. It is a shame to have well diffracting crystals and then throw away good data that could be very useful in refinement. I suggest using $CC1/2$ of around 70% as done for 6E8R and 6EE0 as a cut off. Additionally, the authors should provide Ramachandran plot and rotamer outlier statistics for the structures shown in Table 1. The validation reports for the structures show that some of the PDBs have rather high percentage of rotamer and Ramachandran plot outliers. The authors should carefully examine these in a graphics program such as Coot and correct the structures (given the decent resolution this should really not be a problem).

We thank the reviewer for checking the structural statistics. In the case of 6EDX, despite the strong diffraction the resolution is limited to 2.01 Å by the detector distance that was set during data collection, so the higher resolution data is unfortunately just not available. We have checked structures and now include the Ramachandran statistics as requested.

4) Fig. 1: was care taken to display the pelleting assays using normalized settings? In some of the experiments the background appears very white and in some much darker. Are the different pelleting assays shown in Fig. 1 directly comparable?

*All experiments shown in **Figure 1** were performed using the same concentrations of proteins and lipids, and care was taken to ensure that identical quantities of bound and unbound fractions were loaded for each of the different protein samples. Although the background of the gels can be different from protein to protein as the gels were typically run on different days and stained with different batches of Coomassie stain. Hence in **Figure 2** we ensured that the relative binding of each lipid was normalized based on the ratios of supernatant and pellet fractions so that a direct comparison could be made across all proteins.*

5) For the ITC data shown in Table S3, could the authors please provide standard deviations for the N-values.

The standard deviations for N-values are now provided in an updated Table S3.

6) pg 10, lines 270-272: ‘a certain number of PX domains do bind exclusively to the canonical PtdINS3P...’? Given all the data you present surely you can do better than that.

On page 10, we have altered this sentence to be more explicit inserting the following text: “...we have shown that a significant fraction (12 of 39 tested) of the human PX domains bind exclusively to the canonical PtdIns3P”.

Reviewer #3

The PX domain was first identified as PI3P-binding module almost two decades ago. Numerous biochemical and structural studies have shown since that PX domains can also interact with many other lipids. Now Chandra et. al. are reporting tour de force characterization of 39 of 49 human PX domains by three independent membrane binding analyses. They also performed structural analyses, including determination of a few PX domain structures, structural comparisons, and NMR titration of SNX25-PX-lipid binding, mutational studies, and subcellular localization analysis of selected PX domains in yeast cells. Overall, the work is technically solid and rigorous. It contains the comprehensive and well-organized information about lipid binding properties of PX domains, which would undoubtedly serve as a useful resource for those who study PX domain-containing proteins. The main concern is, however, its incremental nature. This report presents neither a technical breakthrough nor new structural and biological insight. Although the authors should be praised for employing such diverse and orthogonal methods to characterize lipid binding of PX domains, they all represent well established standard methods, hence lack of any new critical structural or functional information. Many previous studies have addressed the physiological significance of diverse lipid specificity of PX domains with in-depth biological studies. Besides classification of PX domains based on their lipid selectivity and some structural rationalization, however, this work does not add much to our understanding of biological insight into how PX domains mediate diverse cellular functions.

We thank the reviewer for the positive comments about the breadth and technical rigor of our work. Reviewer 3 does not ask for any technical, experimental or textual changes. As the editor has stated that they do not share the reviewer’s concern the work may be too incremental in nature, we simply reiterate that we

believe this work will be a major resource for cell and molecular biologists studying the processes of membrane trafficking and cell signaling mediated by peripheral membrane proteins.

Reviewers' Comments:

Reviewer #1:

Remarks to the Author:

The authors have well addressed all previous comments.

Reviewer #2:

Remarks to the Author:

I think that the authors should include the attachment of SEC profiles into the supplemental figures as it goes well with the other figures and there is quite some variability in the quality of the profiles that could be useful in assessing the importance of the other presented data.

Otherwise, the authors have adequately addressed my comments and I recommend publishing the paper.

Revision of manuscript NCOMMS-18-33397

Rev#1

The authors have well addressed all previous comments.

Nothing to address.

Rev#2

I think that the authors should include the attachment of SEC profiles into the supplemental figures as it goes well with the other figures and there is quite some variability in the quality of the profiles that could be useful in assessing the importance of the other presented data. Otherwise, the authors have adequately addressed my comments and I recommend publishing the paper.

This has now been included as Supplementary Fig. S1, and other supplementary figures renumbered accordingly.